# Calcium Signaling Pathways: Key Pathways in the Regulation of Obesity

**DOI:** 10.3390/ijms20112768

**Published:** 2019-06-05

**Authors:** Ziguo Song, Yu Wang, Fei Zhang, Fangyao Yao, Chao Sun

**Affiliations:** College of Animal Science and Technology, Northwest A&F University, Yangling 712100, China; szgken@126.com (Z.S.); wangyu1484@163.com (Y.W.); zhangfei9808@163.com (F.Z.); yaofangyao@163.com (F.Y.)

**Keywords:** calcium signaling pathways, obesity, biological clock, intestinal microbial activity, nervous system excitability

## Abstract

Nowadays, high epidemic obesity-triggered hypertension and diabetes seriously damage social public health. There is now a general consensus that the body’s fat content exceeding a certain threshold can lead to obesity. Calcium ion is one of the most abundant ions in the human body. A large number of studies have shown that calcium signaling could play a major role in increasing energy consumption by enhancing the metabolism and the differentiation of adipocytes and reducing food intake through regulating neuronal excitability, thereby effectively decreasing the occurrence of obesity. In this paper, we review multiple calcium signaling pathways, including the IP3 (inositol 1,4,5-trisphosphate)-Ca^2+^ (calcium ion) pathway, the p38-MAPK (mitogen-activated protein kinase) pathway, and the calmodulin binding pathway, which are involved in biological clock, intestinal microbial activity, and nerve excitability to regulate food intake, metabolism, and differentiation of adipocytes in mammals, resulting in the improvement of obesity.

## 1. Introduction

As an important signaling molecule in cells, calcium ion (Ca^2+^) participates in regulating some important physiological activities in the human body, including the nervous system excitability, the contraction of muscles, the intestinal microbial activity, the activity of enzymes, and the biological clock [1,2]. The concentration of Ca^2+^ remains stable all the time in the matrix, which is tightly bound to the regulation of many different kinds of calcium signaling pathways in the body, such as the IP3 (inositol 1, 4, 5-trisphosphate)-Ca^2+^ pathway, the p38-MAPK (mitogen-activated protein kinase) pathway, and the calmodulin binding pathway.

Adipose tissue is an important energy storage material in animals. Containing an amount of adipose tissue in the body helps maintain normal life activities. On the contrary, accumulating excessive adipose tissue may cause obesity and other diseases [3]. If we can know the regulation mechanism between calcium ions and adipose, it will help to provide a new possibility for decreasing the incidence of obesity. At present, many studies have started to explore the regulation mechanism. They found that calcium ions could be important in enhancing the energy consumption by promoting adipocytes differentiation and metabolism, thereby reducing obesity [4,5,6,7]. In addition to promoting energy consumption, signal molecules such as Leptin, Neuropeptide Y, and Nesfatin-1 could regulate the feeding center and the peripheral satiety system through calcium signaling to decrease food intake and reduce the energy acquisition, thereby decreasing obesity [8,9]. Here, we review the effects of the biological clock, the intestinal microbial activity, and the nerve excitability on the regulation of the incidence of obesity, highlighting the role of multiple calcium signaling pathways in these regulation processes.

## 2. Brief Summary of Calcium Signaling Pathways Related to the Regulation of Obesity

Intracellular calcium ions mainly come from the influx of extracellular calcium ions and the release of calcium ions that are stored in the endoplasmic reticulum. A great number of experimental results indicate that the regulation of intracellular calcium ions concentration involves multiple signaling pathways. In this part, we briefly review the IP3-Ca^2+^ pathway, the p38-MAPK pathway, and calmodulin, which are associated with obesity regulation. 

### 2.1. p38-MAPK Signaling Pathway in the Regulation of Adipocytes Metabolism (Table 1)

During the study of the effects of hypertonic environments on fungi, Brewster et al. [10] found the p38-MAPK signaling pathway for the first time. Five p38-MAPK isoforms are currently known: p38α (p38), p38β1, p38β2, p38γ, and p38δ. The p38-MAPK signaling pathway is mainly composed of a tertiary kinase chain, which is composed of upstream activators such as MKK3, MKK4, and MKK6. MKK3 and MKK6 can specifically activate p38, while MKK4 is different (Figure 1) [11]. Many studies have proven that the p38-MAPK signaling pathway is associated with the concentration of intracellular calcium ions, which might influence the metabolism of adipocytes [12,13]. By stimulating preadipocytes with a p38 MAPK inhibitor and detecting intracellular lipid accumulation, Wang Li et al. [14] demonstrated that intracellular calcium transport in adipocytes could be promoted by activating the p38-MAPK signaling pathway. This mechanism could inhibit lipid accumulation to reduce the incidence of obesity. During Kohlie R’s [15] experiment, they treated murine brown adipocyte cells with dopamine and detected oxygen consumption rates and uncoupling protein 1 levels. The result of this experiment indicated that dopamine directly increased mitochondrial mass and thermogenesis in brown fat through the p38/MAPK pathway. In addition to increasing the adipocytes metabolism, Xie X [16] also reported that miR-21a-5p reduced the differentiation of adipocytes to decrease the occurrence of obesity by inhibiting BPA (bisphenol A) through the p38-MAPK signaling pathway.

### 2.2. Function of Calmodulin in Adipocytes (Table 1)

Calmodulin is a calcium-binding protein that was first discovered in the brain [17]. From that time on, many researchers began to look into its function in the body. Some studies found that the activity of calmodulin could be regulated by affecting the change of intracellular Ca^2+^ concentration, and the function of calmodulin might be related to the body’s metabolic process. Wang Y et al. [18] reported that the Ca^2+^-dependent CaMKK (calcium calmodulin kinase kinase) signaling pathway is involved in WSF-P-1-induced AMPK (adenosine 5‘-monophosphate-activated protein kinase) activation, which could promote adiponectin multimerization in 3T3-L1 adipocytes. By incubating Ca^2+^-deficient tissue in Ca^2+^-free ringer solution, the lipolysis induced by epinephrine, norepinephrine, caffeine, and ACTH (adrenocorticotropin) was depressed. According to the above results, Izawa T et al. [19] suggested that Ca^2+^ might be important in enhancing lipolysis, and calmodulin might promote lipolysis. In addition to regulating lipolysis, Zhang CS et al. [20] demonstrated that CaMKK2 increased energy metabolism of the body through the regulation of the AMPK pathway to decrease the occurrence of obesity. In addition to regulating metabolism-related functions, it has become a common view that the activation of CaMKK2 in hypothalamic neurons could also effectively regulate the feeding behavior of the body to reduce obesity [21]. As for how calmodulin achieves its regulatory function in the body, previous studies proved the specific regulatory mechanism of calmodulin. In this part, we only summarize the mechanism briefly (Figure 1). Calmodulin is semi-activated after the binding of Ca^2+^ and is fully activated after phosphorylation. The activated Ca^2+^·CaM complex can regulate the metabolic process of the body through the action of enzymes corresponding to phosphodiesterase and protein kinase. Calmodulin itself has no enzyme activity. After its activation, it binds to short peptide sequences of corresponding proteins, which can induce changes in its own structure and increase its activity. It also activates the protein by changing its conformation. Changes in Ca^2+^ concentration first cause a change in the conformation and the activity of calmodulin and then activate a target protein containing a CaM binding site, such as a Ca^2+^/CaM-dependent protein kinase [22,23,24]. Therefore, the activity of calmodulin affects the body’s metabolic process by altering changes in intracellular Ca^2+^ concentration.

### 2.3. IP3-Ca^2+^ Signaling Pathway Is Associated with Lipolysis (Table 1)

It is well known that the IP3 pathway is important and can release endoplasmic reticulum calcium to change intracellular calcium concentration. At present, much work has been done to show that the increase of intracellular calcium concentration mediated by the IP3 pathway is related to lipolysis. In Aveseh M’s study [25], they injected calcitonin gene-related peptides into rats intravenously and found that the calcitonin gene-related peptide could promote fat lipolysis through the IP3-Ca^2+^ signaling pathway. By knocking down IP3R (inositol 1,4,5-trisphosphate receptor) in adult drosophila, Subramanian M et al. [26] demonstrated that the loss of the IP3 receptor function in neuropeptide secreting neurons could cause obesity, which is associated with the dysregulation of lipid metabolism. This result means the intracellular calcium signaling pathway (IP3-Ca^2+^) in peptidergic neurons might reduce the occurrence of obesity by promoting lipid metabolism in adult drosophila. For the regulation mechanism of the IP3 pathway, we only make a brief summary in this part (Figure 1). The binding of extracellular signal molecules to GPCR (guanosine-binding protein coupled receptor) could activate the G protein (mainly Go alpha or Gq alpha). In turn, this activated PLC (phospholipase C) to catalyze the hydrolysis of PIP2 (phosphatidylinositol biphosphate) to form IP3 and DAG (diacyl glycerol). Opening of the corresponding calcium channel induced calcium ions entering the cytoplasmic matrix by the binding of IP3 to a specific receptor on the endoplasmic reticulum membrane [27,28]. However, the increase in calcium ion levels mediated by IP3 was only transient, as the activation of the calcium ion pump on the plasma membrane and the endoplasmic reticulum membrane would pump calcium ions out of the cells or into the lumen of the endoplasmic reticulum. On the other hand, an increase of the calcium concentration in the cytoplasmic matrix could reduce the affinity of the channel receptor for IP3 and inhibit IP3-induced calcium ions influx [29,30]. In addition to participating in the metabolism of lipids, several studies have revealed that calcium signaling could also decrease the accumulation of adipose tissue to reduce the occurrence of obesity. Lv bin et al. [31] treated mice with the IP3 pathway excitant, NE (noradrenaline), and the inhibitor, heparin, respectively. In this experiment, they found that the activation of the IP3 pathway could lead to the increase of intracellular calcium ions concentration in white adipose tissue, leading to the increase of adipose accumulation. However, the inhibition of the IP3 pathway resulted in a decrease in intracellular calcium concentration.

### 2.4. Other Calcium Signaling Pathways (Table 1)

In addition to the IP3 pathway, the Rya (ryanodine) receptor channel was also able to release calcium ions in the ER (endoplasmic reticulum) when it was stimulated by some small molecules (NO, caffeine, etc.), thus increasing intracellular calcium ions concentration. This could change neuronal excitability to control the metabolism and the differentiation of adipocytes to reduce the incidence of obesity [32]. Numerous studies have explained that the RyR (ryanodine receptor) pathway and the IP3 pathway could also interact to mediate intracellular transient calcium signaling [33]. Another important signaling pathway for the formation of intracellular calcium signals is extracellular calcium influx. Changes in membrane potential can open the voltage-dependent calcium channels, allowing extracellular calcium ions influx into cells, including L-type, N-type, and T-type calcium channels [34]. In the cAMP-PKA pathway, the effect of PKA (protein kinase A) could be strengthened by the increase of calcium ions concentration, thus affecting the metabolism and the differentiation of adipocytes [35]. Some current studies also found that the Wnt-Ca^2+^ signaling pathway might be related to the altered adipocyte cellularity [36]. The binding of the Wnt protein with crimp Fzd (frizzled) activated the G protein and PLC. This increased calcium ions concentration in the cell and activated PKC (protein kinase C) and CaMKII [36].

**Table 1 ijms-20-02768-t001:** Brief introduction of calcium signaling pathways.

Signaling Pathways	Pathway Composition	Regulation Function	Ref.
P38-MAPK signaling pathway	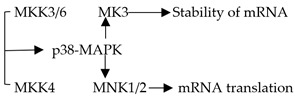	The activation of p38-MAPK signaling pathway in adipocytes can promote intracellular calcium transport to regulate adipocytes metabolism to reduce obesity.	[11,12,13,14,15]
Calmodulin	Calmodulin is semi-activated after binding of Ca^2+^ and is fully activated after phosphorylation. After its activation, it binds to a short peptide, which induces changes in its own structure and increases its activity. It also activates the protein by changing its conformation.	Adipocytes acted by Calmodulin affect energy metabolism to reduce obesity. The activation of CaMKK2 in hypothalamic neurons can regulate the feeding behavior to reduce obesity.	[19,20,21,22,23,24]
IP3-Ca^2+^ signaling pathway	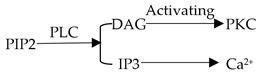	The activation of IP3 pathway can lead to the increase of intracellular calcium ions concentration in adipocytes to regulate lipolysis and the accumulation of adipose.	[27,28,29,30,31]
Other calcium signaling pathways	Rya-Ca^2+^ signaling pathway; L-type, N-type, and T-type calcium channel; cAMP-PKA signaling pathway; Wnt-Ca^2+^ signaling pathway.	Rya receptor channel can release calcium ions in ER/SR to regulate neuronal excitability to regulate the energy metabolism. The opening of voltage-dependent calcium channels can cause extracellular calcium ions influx into cells and regulate the effect of PKA to influence obesity. Wnt-Ca^2+^ signaling pathway can activate PKC and CaMK2 to reduce obesity through increasing intracellular calcium concentration.	[33,34,35,36]

Note: This table summarize the components and functions of the calcium signaling pathways we mentioned. (2.1–2.4). *IP3: inositol 1, 4, 5-trisphosphate; Ca^2+^: calcium ion; p38-MAPK: mitogen-activated protein kinase; Rya: ryanodine; cAMP: cyclic AMP; PKA: protein kinase A; ER: endoplasmic reticulum; CaMK2: calcium calmodulin kinase kinase 2.

## 3. The Nervous System Regulates the Occurrence of Obesity through Calcium Signaling Pathways 

### 3.1. The Nervous System Affects Energy Acquisition by Calcium Signaling Pathways to Decrease the Incidence of Obesity (Table 2)

Obesity is a manifestation of energy imbalance. The imbalance of energy acquisition and energy consumption can affect the occurrence of obesity. Obtaining more energy than consumption can lead to obesity. In recent years, different researchers proved that the mechanism of eating termination was closely related to the occurrence of obesity. When the termination mechanism was easy to start, the food intake was small. On the contrary, the increases of food intake caused obesity [37,38]. Since satiety is important in the feeding termination mechanism, many researchers have supposed that the nervous system might play a significant role in the regulation of the feeding process. Some researchers have shown that the animal feeding process is inseparable from the regulation of the central feeding system and the peripheral satiety system [39]. The results of many studies have suggested that a variety of signaling molecules are involved in feeding regulation, including CCK (cholecystokinin), LP (leptin), NPY (neuropeptide Y), and Nesfatin-1 [8,9]. In this part, we mainly summarize the regulation of Nesfatin-1, which may decrease the occurrence of obesity through calcium signaling pathways. Initially, researchers found that Nesfatin-1 is only distributed in the hypothalamus, but subsequent studies reported that it is ubiquitous in the entire nervous system, while Kim J. [40] also detected NUCB2 (Nucleobindin-2) mRNA expression in adipose tissue. Nowadays, many studies have found that the expression of NUCB2 mRNA can be detected in many peripheral tissues [41,42,43,44]. Reiko et al. [45] demonstrated that Nesfatin-1 expressed in brainstem neurons could enhance the effects of CCK and leptin to reduce food intake. In addition to this study, different studies have reported that Nesfatin-1 expressed in peripheral tissues could affect food intake by directly entering the brain to affect the vagus nerve or the endocrine [46,47]. In most recent studies, it was proven that adipocytes themselves could regulate gastrointestinal motility to reduce food intake by affecting the expression of Nesfatin-1 [48]. As for the relationship with the sympathetic nerve and the calcium signaling pathways, Tanida et al. [49] reported that Nesfatin-1 had a certain regulatory effect on the sympathetic nerve by the PVN (paraventricular nucleus of the hypothalamus) microinjection of Nesfatin-1 in rats. After stimulating the sympathetic nerve in mice, Samantha et al. [50] used voltage and calcium sensitive indicators to detect the change and found that calcium signaling pathways could affect sympathetic nerve excitability. Therefore, many researchers have supposed that rising rates of obesity might occur as a result of Nesfatin-1 influencing calcium signaling pathways.

Nowadays, there is a large volume of published studies describing how Nesfatin-1 decreases the occurrence of obesity through calcium signaling pathways. Christine et al. [51] suggested that the increase in intracellular calcium ions concentration is one of the signs reflecting neuronal activity. During Gantulga’s study [52], they showed that the use of insulin activated Nesfatin-1 neurons and inhibited the feeding process, which decreased the occurrence of obesity. The regulatory mechanism of this process is to increase the calcium ions concentration of Nesfatin-1 neurons in the PVN by promoting extracellular calcium signaling pathways. During more experiments on Nesfatin-1 in PVN, Stengel et al. [53] confirmed that the expression quantity of Nucleobindin-2 mRNA and Nesfatin-1 in PVN were evidently decreased in adult rats after 24 h fasting, which proved that Nesfatin-1 in PVN could inhibit the animal feeding process and reduce the incidence of obesity. Price et al. [54] directly demonstrated that Nesfatin-1 could hyperpolarize or depolarize many neurons in PVN. Another experiment by Price [55] reported that Nesfatin-1 could inhibit food intake and avoid obesity by inhibiting NPY neurons through calcium signaling pathways. Brailou et al. [56] reported that the use of the G-protein receptor inhibitor and the regulation of the L-type calcium channel could reduce the calcium ions concentration in the hypothalamic neurons by inhibiting the regulation of Nesfatin-1 on calcium signaling pathways. After combining Brailou’s results, Price proposed that the interaction of Nesfatin-1 and the G-protein coupled receptors in the hypothalamus could promote calcium ions influx and excite neurons to inhibit the feeding process of animals by enhancing the opening of L-type calcium channels or activating the PKA signaling pathway to reduce obesity. In addition to the study of Nesfatin-1 in PVN, Maejima et al. [57] showed that the neurons associated with Nesfatin-1 in PVN are also associated with other brain functional areas, such as DVC (dorsal vagal complex) and ARC (arcuate nucleus). Moreover, recent studies have found how Nesfatin-1 regulates the calcium signaling pathways in other brain functional areas. For example, Nesfatin-1 can activate the PKC signaling pathway in DVA and affect the excitability of neurons to decrease food intake by increasing the influx of calcium ions [58]. By injecting Nesfatin-1 into VTA (ventral tegmental area), Xi Chen [59] found that Nesfatin-1 could also affect food intake by acting on dopamine neurons.

For Nesfatin-1 expressed in peripheral tissues, extensive work has been done to explore the relationship between obesity and calcium signaling pathways. Iwasaki et al. [60] reported that the feeding process in mice was significantly inhibited by intraperitoneal injection of Nesfatin-1, and they suggested that the N-type calcium channel might take part in the transmission of the signal to the brain by Nesfatin-1 in peripheral tissues to inhibit the process of feeding. To further determine whether N-type calcium channels are associated with Nesfatin-1, many studies have been done in this area. Nesfatin-1 in the peripheral tissues induces calcium ions influx in the vagal afferent nerve through the N-type calcium channel to excite the vagal afferent nerve, and the signal is transmitted to the NTS (nucleus of the solitary tract) by nicotinic cholinergic receptor neurons to inhibit the process of feeding [61,62,63]. As with the N-type calcium channels, researchers have done a great deal of work to explore the relationship between T-type calcium channels and Nesfatin-1. By the injection of Nesfatin-1 in the hindbrain, Ze-Feng Xia et al. [64] found that the T-type calcium channel induced calcium ions influx in DMNV (dorsal motor nucleus of the vagus) neurons, which inhibited the process of feeding by exciting the vagus nerve in the body and inhibiting gastric acid secretion to reduce obesity. At present, the design of many studies increasing the concentration of Nesfatin-1 in the body is based on direct injection in the brain. Hossein Shirvani et al. [65] found that exercise or supplementation of flaxseed oil could also increase the level of Nesfatin-1 in the body, which reduces the food intake through calcium signaling pathways, thereby decreasing the occurrence of obesity. Aside from regulating food intake, many researchers started to focus on the relationship between brown adipose and Nesfatin-1. Yuexin Wang et al. [66] used Nesfatin-1 to treat brown adipocytes directly. They reported that it could promote the metabolism and the differentiation of brown adipocyte cells. By injecting Nesfatin-1 into the ventricles, Yuan et al. [67] found that expressions of UCP-1 (uncoupling protein 1) and intracellular calcium ions concentration were increased in brown adipocyte cells. Therefore, many studies suggested that Nesfatin-1 might regulate the metabolism of brown adipocyte cells through some calcium signaling pathways, such as the AMPK signaling pathway, to increase the energy metabolism of the body, thus reducing the occurrence of obesity [42,68].

### 3.2. The Nervous System Enhances Energy Consumption through Calcium Signaling Pathways to Reduce the Occurrence of Obesity (Table 2)

In addition to controlling energy acquisition by regulating the feeding process to reduce obesity, it is well accepted that the nervous system is involved in regulating adipocytes differentiation and metabolism [69,70,71]. For the first time, Rothwell et al. [72] proved that there is an inhibitory down-regulation pathway between the pons and the medulla that can regulate brown adipocyte cells metabolism. Morrison et al. [73,74] reported that the nucleus pallidus is an important component of the sympathetic nerve that can regulate the heat production of brown adipocytes. By removing FASN (fatty acid synthase) in mouse adipocytes, the activation of the sympathetic innervation was enhanced, and the expression of uncoupling protein 1 in brown adipose also increased rapidly [75]. A great deal of work has been done to confirm that calcium signaling pathways participate in the effect of the sympathetic nerve in regulating adipose metabolism. Researchers found that the metabolic disorders of adipose and the occurrence of obesity are associated with calcium signaling pathways by performing the global transcriptome analysis of brown adipose tissue in obese mice [76]. By using I-mIBG imaging, some researchers indicated that the sympathetic nerve might regulate the metabolism of brown adipose through calcium signaling pathways [77,78].

As for what kinds of calcium signaling pathways are involved in this regulation process, Paulo et al. [79] confirmed that the sympathetic nerve promoted heat production of brown adipose through the cAMP-PKA pathway, which reduced the occurrence of obesity. Tao-Rong et al. [80] reported that the stimulation of the sympathetic nerve could increase the intracellular calcium ions concentration to promote the opening of calcium signaling pathways in mitochondria, which enhanced the mitochondrial function in brown adipocyte cells and affected the ATP consumption to reduce obesity. J Farjam et al. [81] treated adipocytes with different concentrations of calcium ions, observed the morphology of adipocytes by using oil red O, and analyzed GAPDH (glyceraldehyde-3-phosphate dehydrogenase) activity and PPARγ2 gene expression in adipocytes. They found that the opening of calcium channels in the cell membrane resulted in an increase in the concentration of intracellular calcium ions, which could lead to the inhibition of the generation of adipose and consequently reduce the occurrence of obesity. After knocking out or over-expressing the Sik2 gene in mice, the concentration of intracellular calcium ions was increased by the activation of the sympathetic nerve, which activated the cAMP-PKA pathway and increased phosphorylation of Sik2 significantly, leading to the increase of UCP-1 expression and enhancing the brown adipose metabolism to reduce the incidence of obesity in mice [79]. In order to ascertain that the opening of the calcium channels was a key regulatory factor of raising the concentration of calcium ions in cells, many researchers have conducted numerous experiments. They found that TRP (transient receptor potential) channels are important in the regulation of adipocytes metabolism [82,83]. TRPV1 (transient receptor potential channel, subfamily V, member 1) is an important voltage-gated calcium channel that can mediate calcium ions influx into cells. It can be expressed in PVN, DMH (dorsomedial hypothalamic), NTS, and LH (lateral hypothalamic). The sympathetic nerve is important in regulating the opening of TRPV1 [84,85,86,87]. In the study of capsaicin, Varghese et al. [88] indicated that capsaicin could activate TRPV1 channels, and Kawada et al. [89] reported that the sympathetic nervous system could also be activated by capsaicin. These results led to an increased interest in the relationship between TRPV1 channels and the sympathetic nervous system. Ryosuke et al. [90] reported that the function of mitochondria in brown adipocyte cells and the opening of TRPV1 channels could be promoted by increasing the amount of capsaicin. At the same time, they also found that the expression of TRPV1 in the hypothalamus decreased in high-fat diet mice, while the expression of TRPV1 increased after increasing the amount of capsaicin in the diet. After that, they treated mice with TRPV1 inhibitors and found that, although capsaicin activated the sympathetic nerve, the metabolism of brown adipocyte cells and the body weight of mice did not change significantly. Except Ryosuke’s finding, Kunitoshi et al. [7,91] found that the sympathetic nervous systems of normal mice and TRPV1KO mice were excited by cold stimulation, but only the normal mice lost weight with the increase of the concentration of calcium ions in brown adipocyte cells. For TRPV1KO mice, there was no significant change in metabolism and body weight. From these findings, it could be found that the excitability of the sympathetic nervous system could stimulate the opening of TRPV1 channels, which could promote calcium ions influx into adipocytes and enhance the metabolism of brown adipose, thereby decreasing the incidence of obesity. Of course, many researchers proposed that TRPV1 channels might not only be regulated by the sympathetic nerve—it might also be involved in regulating sympathetic excitability. Mazher et al. [92] stimulated the TRPV1 channels of the vagus nerve in NTS or inhibited the TRPV1 channels in advance, and they found that the TRPV1 channel of the vagus nerve in NTS could be involved in inhibiting the sympathetic nerve excitability of brown adipocyte cells, which could attenuate the metabolism of adipocytes and cause obesity in mice. Up to now, a considerable amount of studies have been conducted on this problem. The nerve signal transmitted by the vagus nerve inhibits the sympathetic nerve’s strengthening effect on the metabolism of brown adipocyte cells, thus regulating the calcium channels in vagus nerve could regulate the body’s energy metabolism to reduce the occurrence of obesity in mice [93,94]. EPA (eicosatetraenoic acid) and DHA (docosahexaenoic acid) in fish oil could stimulate the vagus nerve in the intestine through the TRPV1 channels to regulate the development of the sympathetic nerve, thereby reducing the occurrence of obesity [95].

In addition to TRPV1 channels, researchers also studied another calcium ion channel, TRPA1 (transient receptor potential A1). Yokoyama’s electrophysiological experiments in the suprachiasmatic nucleus of the hypothalamus suggested that TRPA1 channel is important in the regulation of the excitability of nerves [96]. By treating mice with TRPA1 stimulant, cinnamaldehyde, Zuo et al. [97] found that cinnamaldehyde could promote white adipose browning to prevent diet-induced obesity. In addition to regulating extracellular calcium ions influx, the regulation of the release of calcium ions in the endoplasmic reticulum of the cell through the IP3-DAG (diacyl glycerol) pathway might also affect calcium homeostasis in adipocytes. IP3R and RyR are calcium channel receptors on the endoplasmic reticulum. Junfeng Bi et al. [98] reported that these receptors regulated the calcium homeostasis of adipocytes and decreased the accumulation of fat droplets in adipocytes to reduce the occurrence of obesity. Bernhard et al. [99] showed that Ach (acetyl choline) released by the sympathetic nerve could regulate the release of calcium ions in the endoplasmic reticulum through the Gαq/PLC/IP3 pathway to increase intracellular calcium ions concentration. In addition to the study of calcium channels, Mate Maus et al. [100] found that SOCE (store-operated Ca^2+^ entry), which is regulated by STIM1 (stromal interaction molecule 1), could increase intracellular calcium ions concentration and affect lipid droplet accumulation in adipocytes. Nowadays, many researchers have found that the new pathway formed by the cross-talk between the calcium signal and other signaling molecules can also regulate the nervous system to affect the metabolism and the differentiation of adipocytes to influence the incidence of obesity. By activating or knocking out the KCNK3 gene in mice, Yi Chen et al. [101] found that the potassium outflow caused by the activation of KCNK3 inhibited the calcium ions influx through VDCCs (voltage dependent calcium channels) and decreased the heat production of brown adipocyte cells to reduce obesity in mice—the result in knockout mice was the opposite. At the same time, they also found that the activation of KCNK3 is regulated by the nervous system. It is also one of the current research directions using the functional substances contained in food to regulate the excitability of the sympathetic nerve through calcium signaling pathways to enhance the metabolism of adipocytes, thereby reducing the occurrence of obesity [38]. For example, taurine, olive bitters, and capsaicin present in fish oil and olive oil could reduce the occurrence of obesity [102,103,104].

**Table 2 ijms-20-02768-t002:** The summary of the main calcium signaling pathways involved in the regulation of obesity by the nervous system.

Target	Signaling Pathways	Regulatory Region	Regulation Process	Ref.
Decreasing Food Intake	L-type calcium channel	PVN	Inhibiting the opening of L-type calcium channels can decrease calcium ions influx to reduce food intake.	[54,55,56]
	PKC signaling pathway	DRG (dorsal root ganglion)	The activation of PKC signaling pathway can affect the excitability of neurons to decrease food intake.	[57,58]
	N-type calcium channels	Vagal Afferent Nerve	N-type calcium channels can excite vagal afferent nerve, and the signal is transmitted to the NTS (nucleus of the solitary tract) to inhibit food intake.	[61,62,63]
	T-type calcium channels	DMNV	T-type calcium channels can induce calcium ions influx in DMNV, which can inhibit food intake and gastric acid secretion by exciting the vagus nerve.	[64]
Increasing the Energy Metabolism	AMPK signaling pathway	Brown Adipose	Increasing the expression of UCP-1 to enhance the brown adipose metabolism.	[42,66,67,68]
	CAMP-PKA signaling pathway	Brown Adipose	Sympathetic nerve increases the expression of UCP1 to promote heat production in brown adipose by cAMP-PKA pathway.	[79]
	TRPV1 channel	Brown Adipose	Sympathetic nervous system can promote the opening of TRPV1 channel to enhance the heat production of brown adipose.	[7,90,91]
		Vagus Nerve in NTS	TRPV1 channel can inhibit the sympathetic nerve excitability of brown adipose.	[92,93,94,95]
	TRPA1 channel	White Adipose	PYY activates the IP3-DAG-Ca^2+^ signaling pathway to inhibit the apoptosis of islet cells.	[96,97]
	IP3R/RyR	Adipocytes	Promoting the opening of IP3R and RyR can regulate the calcium homeostasis of adipocytes and decrease the accumulation of fat droplets.	[98,99]

PVN: paraventricular nucleus of hypothalamus; PKC: protein kinase C; PYY: peptide YY; RyR: ryanodine receptor; AMPK: adenosine 5‘-monophosphate-activated protein kinase; TRPV1: transient receptor potential channel, subfamily V, member 1; TRPA1: transient receptor potential A1; UCP-1: uncoupling protein 1; DMNV: dorsal motor nucleus of the vagus; DAG: diacyl glycerol; DRG: dorsal root ganglion; NTS: nucleus of the solitary tract.

## 4. The Biological Clock and Intestinal Microbial Regulate Obesity by Influencing Energy Acquisition and Consumption through Calcium Signaling Pathways 

### 4.1. The Biological Clock Acts on the Central Nervous System through the Calcium Signal Pathways to Regulate Energy Consumption and Energy Acquisition, thus Affecting the Occurrence of Obesity (Table 3)

The daily oscillations of the mammalian rhythm are controlled by the intrinsic circadian clock in the SCN (supranational nucleus) of the hypothalamus [105,106]. The SCN is the main circadian rhythm pacemaker in the body, and resynchronization of the body clock is achieved principally through integrating input from the environment, such as light exposure, to align the physiological and the behavioral rhythms of organisms [107]. The external light affects animal feeding behavior, sleep behavior, and body energy metabolism by affecting the animal’s biological clock through the SCN. The SCN affects calcium signaling pathways through the LD cycle (light/dark cycle), which could change the content and the activity of the biological clock protein to control the expression level of the lipid synthesis gene in the peripheral tissues and reduce the accumulation of lipids in the body, thereby decreasing the occurrence of obesity [108,109]. Burkeen et al. reported that calcium signaling pathways in the mitochondria of adipocytes could mediate the accumulation of extracellular ATP through SCN rhythm, which affects the body’s energy metabolism, decreasing the occurrence of obesity [110,111]. Calcium signaling pathways are also engaged in the activation of daily neurons in the SCN. There is an ongoing circadian rhythm oscillation in the membrane potential of neurons, and the peak midday rhythm is a direct result of the calcium signaling pathways in neurons [112,113]. It has been found that CaCCs (calcium-regulated chloride channels) could be present in mammalian epithelial cells and excitable cells. By immunohistochemistry and Western blotting, Aguilar-Roblero et al. found that the expression of the CaCCs anoctamin-1 and intracellular calcium concentration in internal sediments increased, which proved that CaCCs might be present in SCN neurons and involved in the regulation of nerve cell excitability [114,115]. Rya is part of the biological clock output pathway. Changes in intracellular and extracellular calcium levels caused by Rya could directly affect neuronal activity to alter the energy harvesting and the depletion process in the body to reduce obesity [115,116,117].

Changes of calcium signaling pathways induced by night light could lead to changes in adipogenic protein expression and cause the occurrence of fatty degeneration. Exposure to ANAL (artificial light at night) could generate obesity, hyperlipidemia, and reduce the efficiency of insulin in promoting glucose uptake and utilization. Some researchers have found that exposure to light at night is associated with obesity and other diabetes by regulating the biological clock [115,118]. For most organisms, the day is characterized by two distinct stages of behavior—one stage with active feeding behavior, and another stage with resting/sleeping and fasting behavior. During the active period, the body controls nutrient intake through calcium signaling pathways. Resulting from the influence of nighttime illumination on the hypothalamus, the regulation of calcium signaling pathways causes the organism to consume more energy to reduce the incidence of obesity [119,120]. During the rest, the stored energy maintains normal metabolism, and some researchers believe that night illumination might affect the calcium homeostasis in adipocytes by influencing calcium signaling pathways, which regulate normal energy metabolism and ultimately result in obesity. The biological metabolic process is a function of the LD cycle, which has a distinct day/night rhythm. Continuous light irradiation of mice results in decreased glucose tolerance and destruction of calcium homeostasis in adipocytes, thereby leading to obesity [121,122]. Continuous irradiation of nighttime dim light after a high-fat diet in mice results in a rapid increase in body weight. After the mice return to normal LD cycles, the body weight gain rate also returns to normal, indicating that it is a reversible process that the biological clock causes obesity by leading to metabolic disorders in the body through calcium signaling pathways [123,124]. Exposure to dim light at night and high-fat diet increases body weight, reduces glucose tolerance, and alters insulin secretion as compared to light and dark cycles, and the effects of dim light at night with high-fat diet could be superimposed [124].

### 4.2. The Intestinal Microflora Regulated by the Change of Biological Clock Rhythm Affects Energy Intake and Energy Metabolism through Calcium Signaling Pathway, thus Affecting the Occurrence of Obesity (Table 3)

The SCN, the light signal receiver of the body, receives a circadian signal that changes the body’s circadian rhythm. Changes in the body clock caused by light eliminate the circadian rhythm changes in the composition of intestinal microorganisms by affecting the food intake rhythm [125,126,127]. Wu et al. compared the composition of intestinal microorganisms in mice under the condition of a regular LD cycle and a DD cycle (continuous darkness). They found that the rhythmic changes of intestinal microorganisms in almost all parts of the intestine disappeared in mice under the condition of continuous darkness [128]. This study suggests that photoperiod changes are an upstream factor in the circadian rhythms of gut microbes. The deletion of the Bmal1 gene, the central component of the forward limb in the host circadian clock, made the circadian rhythm of mice’s intestinal microorganisms disappear, and the abundance and the composition of the intestinal microorganisms changed [129]. It was proven that the loss of Per1 and Per2, key genes that regulate the host biological clock, led to the rhythmic loss of intestinal microorganisms [125]. The rhythmic changes of intestinal microorganisms acted on the metabolic activities of the host intestinal epithelial cells to achieve the circadian rhythm of regulating host metabolism [130].

SCFAs (short-chain fatty acids), secreted by gut microbes, regulate energy metabolism, thereby affecting the occurrence of obesity. Prior studies have noted the importance of intestinal microflora, which could regulate obesity. Intestinal microflora play an important role in regulating obesity [131,132,133]. The intestinal microflora produces many metabolites with physiological activity during its activity. Of the key metabolites are SCFAs, including acetate, propionate, and butyric acid, which could regulate energy uptake and metabolism. The species, the quantity, and the source of microorganisms in intestine influence the release of SCFAs [134,135,136]. The current study found that resistant starch increased early-phase insulin, C-peptide, and GLP-1 (glucagon-like peptide-1) secretion [137]. The intestinal microbial diversity of mice fed resistant starch type 3 was lower than that of mice in other groups, which affected the intestinal SCFAs represented by butyrate, thereby increasing the intestinal absorption of calcium, magnesium, and iron [138]. Changes in insulin secretion are closely related to obesity. Propionic acid SCFA increases insulin secretion, which has nothing to do with changes of GLP-1 level, and this mechanism is associated with PKC [139]. An interesting study found that acorn-derived probiotics and Xigu-derived probiotics improved the glucose metabolism deficit by inducing high fat intake through the intestinal microbial-brain axis regulation pathway, thereby preventing or treating diet-induced obesity [140].

GLP-1 secretions of intestinal epithelial L cells regulated by SCFA regulates feeding behavior, metabolism of fat cells, and the survival and the metabolism of pancreatic cells, thereby affecting the occurrence of obesity. Cani et al. [141,142] reported a doubling of GLP-1 at the proximal colon in male Wistar mice treated with fructooligosaccharides. This phenomenon was due to the fact that dietary fiber changed the composition of SCFAs, which up-regulated neurogenin 3 and NeuroD differentiation factors to promote the increase of L cells in the intestinal tract. GLP-1R is the receptor of GLP-1 and is expressed in many tissues, such as pancreas, hypothalamus, and adipose tissue [143,144]. GLP-1 can act on the hypothalamic arcuate nucleus highly expressing GLP-1R to inhibit food intake [145]. Some studies indicate that GLP-1 affects the secretion of Insulin and glucagon through related calcium signaling pathways to influence obesity [146]. It is noteworthy that GLP-1 could act on islet B cells and accelerate its proliferation and prevent its apoptosis; some researchers pointed out that GLP-1 might promote proliferation of islet B cells by activating Akt and PKC signaling pathways (IP3-DAG-Ca^2+^ pathway). However, the anti-apoptotic signaling pathways of islet B cells mediated by GLP-1 have yet to be fully defined. Researchers suggest that calcium signaling pathways might be participating in regulating the anti-apoptotic process [147]. It was also demonstrated that GLP-1 promotes insulin secretion by phosphorylating the PKA signaling pathway’s downstream as the Ca^2+^ sensor, Syt7 [148]. In addition to regulating pancreatic cells, GLP-1 also takes part in adipocytes metabolism. By using the GLP-1R agonist, liraglutide, Beiroa et al. [149] found that it could facilitate brown adipose tissue thermogenesis and adipose browning in mice, which was relative to the hypothalamic ventromedial nucleus and the action of the AMPK pathway. GLP-1 could also directly induce pre-adipocyte proliferation through the RTK-Ras-Raf-MEK-ERK (extracellular signal-related kinase) pathway, the PKC (IP3-DAG-Ca^2+^ pathway), and the AKT signaling pathway to increase the incidence of obesity [144]. Moreover, the use of flavonoid eriodictyol in C57BL/6N mice on a high-fat diet revealed decreased expression of adipogenesis related genes in white adipose tissue and up-regulated the expression of fatty acid oxidation-related enzymes and genes in liver, which might be related to the changes in the production and the release of GLP-1 [150]. Some recent findings showed that a high-fat diet promoted the proliferation of methanogens in the gastrointestinal tract in the process of inducing obesity. The increase of methane in the intestinal tract enhanced the secretion of GLP-1, which could promote the cAMP signal pathway in participating in this process [151]. High-fat and high-sucrose diets stimulated the release of GLP-1 and insulin. These processes regulated postprandial blood sugar levels to maintain normal levels, alleviated glucose-induced intolerance, and reduced body fat synthesis [152].

Secretions of intestinal epithelial L cells regulated by SCFA, PYY1-36, and PYY3-36 regulate feeding behavior, metabolism of fat cells, and survival and metabolism of pancreatic cells, thereby affecting the occurrence of obesity. As a significant material of intestinal endocrine cells, PYY (peptide YY) acts in the regulation of obesity. According to the imaging of living cells, specific expression of fluorescent calcium sensors GCaMP3 of L cells using angiotensin II could improve the L cells in mice and human Ca^2+^ concentrations to stimulate the release of GLP-1 and PYY [153]. SCFAs could increase the density of PYY cells in the colon and the concentration of circulating PYY in blood, acting on the related calcium signaling pathways through FFAR2 (free fatty acid receptor 2) [154,155]. It is a general consensus that PYY is mainly composed of PYY1-36 and PYY3-36, which are mediated by the NPY family of G protein-coupled receptors [156]. PYY activates intracellular calcium signaling pathways by binding to NPY receptors expressed in the nervous system, the pancreas, the intestines, and other tissues, thus realizing the regulation of energy intake and metabolism. PYY1-36 is more important than PYY3-36 in reducing the secretion of insulin caused by the change of glucose concentration, but it does not affect the basic secretion of insulin. PYY3-36 plays a major role in inhibiting food intake and affecting insulin sensitivity [157,158,159,160]. For one thing, PYY3-36 plays an essential role in inhibiting food intake through acting on PVN [161]. For another, the injection of PYY3-36 in insulin-resistant mice increases insulin sensitivity during the hyper insulin-glycemic phase. This mechanism increases the metabolic rate of glucose and prevents obesity [162]. PYY1-36 and PYY3-36 decrease the secretion of insulin in immortalized rodent and human islet B cells isolated mouse islets by impeding alterations in cell membrane potential, Ca^2+^ concentrations, and the elevations of cAMP [163]. As with GLP-1, PYY acts as a regulatory factor involved in the proliferation and the anti-apoptosis of islet cells with the purpose of maintaining insulin homeostasis in the body. These mechanisms might affect the occurrence of obesity. PYY is clearly critical for mitosis of pancreatic islet cells, which activate the phospholipase C related pathway, the mitogen activated protein kinase, and the ERK1/2 phosphorylation to control the expression of proliferation-related genes [164,165,166,167]. In apoptosis, some studies indicate that the expression of genes in the nucleus is regulated by the IP3-DAG-Ca^2+^ signaling pathway, and that the NPY receptor is also involved [168]. In addition, intraperitoneal injection of PYY3-36 or Y2R agonists in mice improves hepato-portal active GLP-1 plasma levels and glucose tolerance to nutritional stimulation, thereby affecting glucose homeostasis [169]. PYY has been recently reported in enhancing lipogenic capacity [170]. It was reported that the use of yacon roots in Wistar male rats on a high-fat diet model increased glucagon and PYY levels, thereby inhibiting lipogenesis [171]. The intake of low-calorie and high-protein food dramatically increased PYY concentration, which reduced appetite through stimulating nervous system exciting, and calcium signaling was involved [172]. Additionally, epigallocatechin-3-gallate changes fatty acid uptake, fat production, and the expression of metabolism related regulatory genes in adipose tissue to reduce the occurrence of obesity, and the increase of GLP-1 and PYY levels might be involved in this regulatory process [173].

**Table 3 ijms-20-02768-t003:** The summary of the main calcium signaling pathways involved in the regulation of obesity by the biological clock and the intestinal microbial.

Target	Signaling Pathways	Regulatory Region	Regulation Process	Ref.
Metabolism, proliferation and apoptosis of adipose tissue	CaCCs/Rya-Ca^2+^ signal pathway	SCN neurons	Activation of the CaCCs/Rya-Ca^2+^ signaling pathway in SCN neurons leads to the accumulation of extracellular ATP in adipocytes.	[110,111,112,113,114,115,116,117]
	AMPK signal pathway	hypothalamus	GLP-1 acts on the AMPK pathway in the hypothalamus, promoting heat production in brown adipose tissue and browning of adipocytes.	[149]
	ERK, PKC and AKT signal pathways	pre-adipocyte	GLP-1 acts on pre-adipocyte to promote its proliferation and inhibit its apoptosis by activating the ERK, PKC and AKT signaling pathways.	[144]
Metabolism, proliferation and apoptosis of islet cells	PKA-syt7 signaling pathway	β-cell	The activation of PKA-sty7 signaling pathway activated by GLP-1 promotes insulin secretion.	[148]
	AKT and PKC signaling pathways	β-cell	GLP-1 activates the AKT and PKC pathways to promote the proliferation of islet B cells.	[147]
	PLC and ERK1/2 signal pathways	islet cells	PYY promotes mitotic proliferation of islet cells by activating the PKC and ERK1/2 signaling pathways.	[164,165,166,167]
	IP3-DAG-Ca^2+^ signal pathway	islet cells	PYY activates the IP3-DAG-Ca^2+^ signaling pathway to inhibit the apoptosis of islet cells.	[168]
	cAMP signal pathway	β-cell	PYY inhibits the increase of alterations cell membrane potential, cAMP and Ca^2+^ concentration to reduce the secretion of insulin.	[163]

CaCCs: calcium-regulated chloride channels; SCN: supranational nucleus; GLP: glucagon-like peptide-1; ERK: extracellular signal-regulated kinase.

## 5. Conclusions and Perspectives

Current studies indicate that obesity is monitored by a variety of factors, whether it is intestinal microbial regulation, biological clock regulation, or central nervous system regulation, which are not separable from the calcium signaling pathways (Figure 2).

In these adjustment modes, the role of the nervous system in regulating calcium signaling pathways is the most important. Regulating calcium signaling pathways can affect the development of the central nervous system and regulate the excitability of feeding neurons in the hypothalamus to affect the feeding process of animals to reduce energy intake, thereby reducing the occurrence of obesity. In addition to reducing energy intake, the nervous system might also increase its heat production in adipocytes through calcium signaling pathways to increase the energy consumption of the body to achieve the purpose of weight loss. The DAG-IP3 signaling pathway and the cAMP-PKA pathway are the major calcium signaling pathways of the nervous system that could regulate the proliferation of adipocytes. Whether it stimulates the opening of calcium channels on the cell membrane to cause calcium ions influx or stimulates the endoplasmic reticulum to cause the release of intracellular calcium ions, once the intracellular calcium ions concentration increases, the cAMP-PKA pathway in brown adipocyte cells is activated to enhance mitochondrial activity in brown adipocyte cells, thereby increasing caloric production and reducing the occurrence of obesity.

The change of the biological clock caused by night illumination also has a definite regulatory effect on the occurrence of obesity through calcium signaling pathways. Continuous light at night can not only affect the hypothalamus through calcium signaling pathways but can also affect feeding behavior. It also causes metabolic disorders in the animal that affect the calcium homeostasis in the neuronal cells; this causes changes in the nervous system excitability, which affects the body’s energy expenditure process in reducing obesity. In addition to the use of night light, there have been many studies that decreased the occurrence of obesity by regulating the time of feeding. These studies have shown that the secretion of SCFAs, insulin, and some brain-gut peptides are affected during this regulation process. We believe that calcium signaling pathways may be a key part in this regulation process and could be a future research direction.

The change of circadian rhythm changes the body’s eating behavior and the expression of genes related to biological rhythm; it then changes the composition of intestinal microflora. SCFAs secreted by intestinal microorganisms are affected by microbial flora structure, food intake, and other factors. GLP-1 and PYY secreted by L cells in the intestinal epithelium are regulated by SCFAs. SCFAs, GLP-1, and PYY influence energy intake and energy metabolism to affect the occurrence of obesity by activating various pathways through calcium signaling pathways, including neuronal excitability, islet cell proliferation and apoptosis, insulin and glucagon secretion, adipose tissue metabolism, and gastrointestinal tract activity. However, what is the specific relationship between SCFAs, GLP-1, and PYY in regulating energy metabolism? Many studies have shown that SCFAs, PYY, and GLP-1 have regulatory effects on the absorption of nutrients in the intestinal tract, thus affecting the occurrence of obesity. We think that calcium signaling pathways may be involved in these regulation paths, and these problems could be used as future research directions.

## Figures and Tables

**Figure 1 ijms-20-02768-f001:**
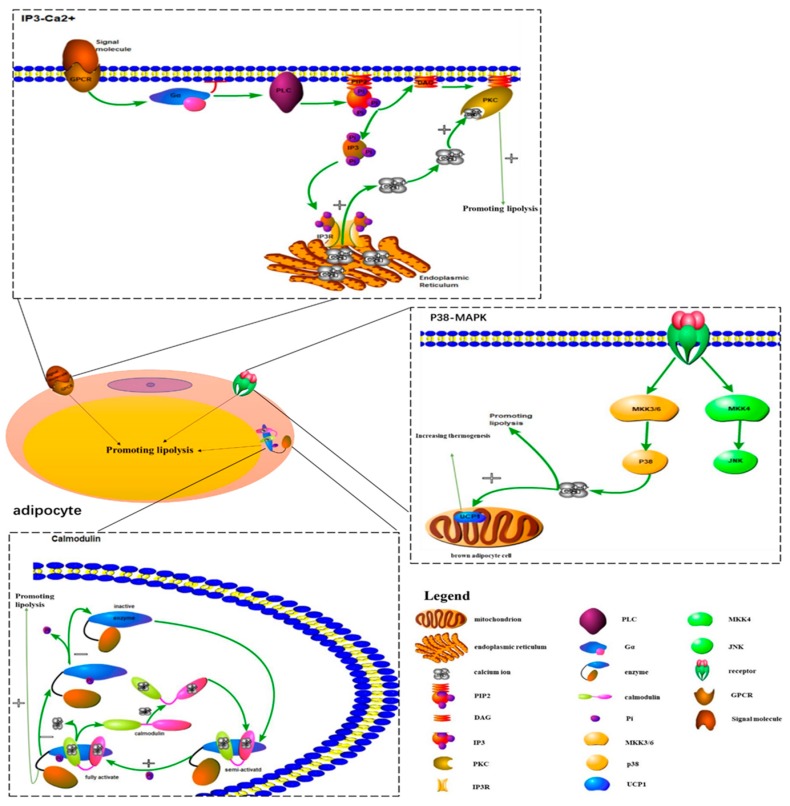
Signal transduction mechanisms of the main calcium signaling pathways related to obesity in adipocytes.

**Figure 2 ijms-20-02768-f002:**
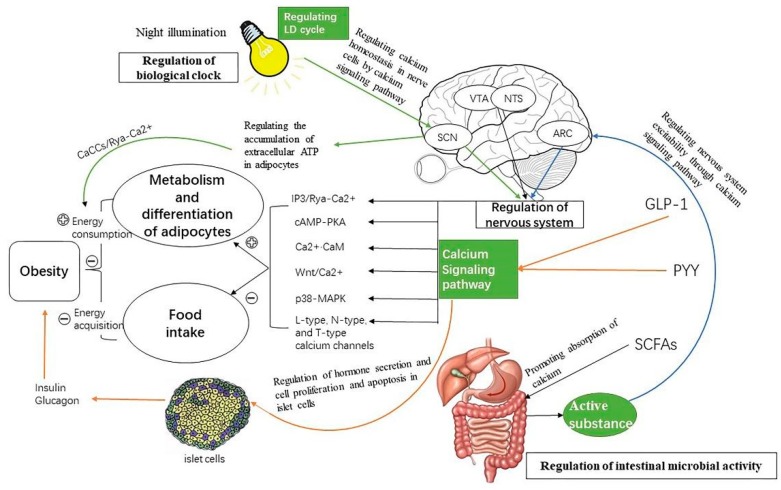
The description of the mechanisms by which the nervous system, the gut microbes, and the circadian rhythms regulate obesity through calcium signaling. This paper summarizes the diagrams of nervous system, light rhythm, and intestinal microorganisms regulating energy consumption and acquisition through calcium signaling pathways. Light changes the activity of SCN neurons by activating calcium signaling pathways, and these signals are transmitted through neurons to brain regions associated with food intake to enhance satiety and inhibit food intake. On the other hand, a signal is transmitted to adipocytes by light, which can promote adipocyte metabolism and heat production by activating DAG-IP3 and cAMP-PKA pathways to increase energy consumption. GLP-1, PYY, and SCFAs secreted by intestinal microorganisms act on the PVN region of the brain, activating feeding-related neurons to enhance satiety and inhibit feeding. IP3-DAG and RTK-Ras-raf-MEK-ERK pathways act on adipocytes to regulate adipocyte metabolism and promote lipid decomposition [25,26]. Directly acting on pancreatic tissue: the proliferation of islet B cells is promoted by many calcium signaling pathways, such as the PLC (phospholipase C)-ERK1/2 and the PKC pathways, and the apoptosis of islet B cells is inhibited by the IP3-DAG and the MAPK pathways to ensure that the pancreatic tissue meets the needs of insulin, which may be related to obesity. Signals regulate insulin secretion through the cAMP and the PKA-Syt7 pathways, which are positively or negatively correlated with obesity. Signaling molecule Nesfatin-1 acts on neurons of VTA (ventral tegmental area) and NTS through N-type and T-type calcium channels, enhancing satiety and inhibiting feeding [55,60,61,62,63,64]. Signals are transmitted to adipocytes when there is sympathetic excitation, and the metabolic heat production of brown adipocytes is regulated by cAMP-PKA, TRPV1, TRPA1, and -IP3-DAG pathways to increase energy consumption [83,92,93,94].

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
