# Peer review of "Calcium Signaling Pathways: Key Pathways in the Regulation of Obesity"

_ijms, 2019, doi:10.3390/ijms20112768_

Reviewer 1 Report

This review entitled “Calcium signaling pathways: key pathways in the regulation of obesity” authored by Song et al. provides a comprehensive story that link the roles of different types of calcium signaling in the regulation of energy metabolism involving the CNS and multiple peripheral systems. This is an important field, but the topic is not so well explored as other topics in the metabolic research. This review would bring enthusiasm into this topic, which is a merit of this review.

 It is not surprising that calcium signaling regulates secretion of almost all water-soluble chemical messengers and serves as a ubiquitous intracellular signaling messengers.  This manuscript covers broad studies. Some parts, for example discuss TRP channels on pages 6-7, provide necessary details and therefore informative. Some parts, however lacks specific details when describe many of these studies or authors’ own perspective and opinions.

 Section 4 includes the effects of biological clock and gut microbiome on metabolic regulation by calcium signaling. These two parts are separately discussed without strong link between them. Maybe re-organize to central regulation (via biological clock) vs peripheral regulation (via gut microbiome).

 The writing should be improved. The misused words (including scientific terms, such as ARC on line 180) and tense of sentences, and confusing statements throughout the manuscript should be rewritten to increase the clarity.  Additionally, do not direct quote in scientific writing (lines 173-175).

Author Response

Dear Reviewer:

On behalf of my co-authors, we thank you very much for giving us an opportunity to revise our manuscript, we are very grateful for reviewer#1’s positive and constructive comments and suggestions on our manuscript entitled “Calcium signaling pathways: key pathways in the regulation of obesity”. These suggestions and comments are valuable and very helpful for revising and improving our paper.

Revised portions are marked in red in the revised paper. The following is a point-to-point response to the reviewer's comments and responses are in red.

 Responses to Reviewer#1

 Point 1. Review Comments: This manuscript covers broad studies. Some parts, for example discuss TRP channels on pages 6-7, provide necessary details and therefore informative. Some parts, however lacks specific details when describe many of these studies or authors’ own perspective and opinions.

 Response 1: Thanks for your suggestion; it is important that we need to give more details to make readers understand these viewpoints clearly. We have given specific details for other sections (section2, section3 and section4) just like the discussion of TRP channels. The form is: The researchers worked out XX conclusions by using XX experimental methods in XX experimental animals or XX cells.

This is an example: During Kohlie R’s experiment, they treated murine brown adipocyte cells with dopamine and detected oxygen consumption rates and uncoupling protein 1 levels. The result of this experiment indicated that dopamine directly increased mitochondrial mass and thermogenesis in brown fat through p38/MAPK pathway.

All the re-write sentences in the revised paper are in the same format as in the example.

 Point 2. Review Comments: Section 4 includes the effects of biological clock and gut microbiome on metabolic regulation by calcium signaling. These two parts are separately discussed without strong link between them. Maybe re-organize to central regulation (via biological clock) vs peripheral regulation (via gut microbiome).

 Response 2: Thanks for your thoughtful comments and suggestions. We have found a new connection between biological clocks and gut microbiota that the change of body clock rhythm can cause the change of circadian rhythm of intestinal microflora by looking up more research papers.

SCN, the light signal receiver of the body, makes the biological clock rhythm of the body change, and the signal is transmitted to the feeding center of the body to regulate the feeding behavior of the body. This change in feeding rhythm alters the composition of intestinal microflora. In addition, the change of biological clock rhythm also causes the expression of related genes to fluctuate, thus affecting the circadian rhythm of intestinal microorganisms. Changes in the composition of the intestinal microbiota can cause fluctuations in the secretion of SCFAs. On the one hand, SCFAs can directly participate in the regulation of energy metabolism in the body. On the other hand, SCFAs can act on L cells in the intestinal epithelium to regulate the synthesis and release of GLP-1 and PYY. Fluctuations in the concentrations of GLP-1 and PYY can regulate energy intake and energy metabolism.

These are the research papers we add in the revised paper:

129. Thaiss CA, Zeevi D, Levy M, Zilberman-Schapira G, Suez J, Tengeler AC, Abramson L, Katz MN, Korem T, Zmora N, et al. Transkingdom control of microbiota diurnal oscillations promotes metabolic homeostasis. Cell. 2014,159, 514-29. [CrossRef] [PubMed]

130. Zarrinpar A, Chaix A, Yooseph S, Panda S. Diet and feeding pattern affect the diurnal dynamics of the gut microbiome. Cell Metab. 2014, 20, 1006-17. [CrossRef] [PubMed]

131. Voigt RM, Forsyth CB, Green SJ, Mutlu E, Engen P, Vitaterna MH, Turek FW, Keshavarzian A. Circadian disorganization alters intestinal microbiota. PLoS One. 2014, 9, e97500. [CrossRef] [PubMed]

132. Wu G, Tang W, He Y, Hu J, Gong S, He Z, Wei G, Lv L, Jiang Y, Zhou H, et al. Light exposure influences the diurnal oscillation of gut microbiota in mice. Biochem Biophys Res Commun. 2018, 501, 16-23. [CrossRef] [PubMed]

133. Liang X, Bushman F D, Fitzgerald G A. Rhythmicity of the intestinal microbiota is regulated by gender and the host circadian clock[J]. Proc Natl Acad Sci U S A. 2015, 112, 10479-84. [CrossRef] [PubMed]

134. Thaiss CA, Zeevi D, Levy M, Zilberman-Schapira G, Suez J, Tengeler AC, Abramson L, Katz MN, Korem T, Zmora N, et al. Transkingdom control of microbiota diurnal oscillations promotes metabolic homeostasis. Cell. 2014, 159, 514-29. [CrossRef] [PubMed]

135. Thaiss CA, Levy M, Korem T, Dohnalová L, Shapiro H, Jaitin DA, David E, Winter DR, Gury-BenAri M, Tatirovsky E, et al. Microbiota diurnal rhythmicity programs host transcriptome oscillations. Cell. 2016, 167, 1495-1510.e12. [CrossRef] [PubMed]

 Point 3. The writing should be improved. The misused words (including scientific terms, such as ARC on line 180) and tense of sentences, and confusing statements throughout the manuscript should be rewritten to increase the clarity. Additionally, do not direct quote in scientific writing (lines 173-175).

 Response 3: We are very sorry for our incorrect and confusing writing, thanks for pointing out our mistakes. We have corrected the misused words (such as ARC, we have made a correction--Arcuate Nucleus) and revised the problem of the direct quote in our paper (The new sentence--Brailou et al. reported that the use of the G-protein receptor inhibitor and the regulation of the L-type calcium channel could reduce the calcium ions concentration in the hypothalamic neurons by inhibiting the regulation of Nesfatin-1 on calcium signaling pathways). In addition to correcting these errors mentioned above, we have simplified the sentence structure, increased multiple types of sentences, and unified the tense (The tense is unified into the past tense). As for the confusing statements, we have rewritten some neutral vocabulary (such as regulate, influence and affect et al.) and done extensive English editing in this revised paper.

 We tried our best to improve the manuscript and made many changes in the manuscript.  These changes will not influence the content and framework of the paper. And here we did not list all the changes but marked in red in our revised paper.

We appreciate for reviewer#1’s warm work earnestly, and hope that the correction will meet with approval.

Once again, thank you very much for your comments and suggestions.

 Yours sincerely,

Ziguo Song

Name: Chao Sun

E-mail: sunchao2775@163.com

Reviewer 2 Report

This review paper summarized calcium pathway and association with obesity. This manuscript is very easy to understand but extra figures/tables may help readers to understand more.

Author Response

    Dear Reviewer:

On behalf of my co-authors, we thank you very much for giving us an opportunity to revise our manuscript, we are very grateful for reviewer#2’s positive and constructive comments and suggestions on our manuscript entitled “Calcium signaling pathways: key pathways in the regulation of obesity”. These suggestions and comments are valuable and very helpful for revising and improving our paper.

Revised portions are marked in red in the revised paper. The following is a point-to-point response to the reviewer's comments and responses are in red.

Responses to Reviewer#2

 Point 1. Review Comments: This manuscript is very easy to understand but extra figures/tables may help readers to understand more.

 Response 1: Thank you very much for your great efforts on our manuscript. It is true that we need more figures and tables to help readers to understand this review easily. In this revised paper we have added a figure to explain the regulatory mechanism of the main calcium signaling pathways in section 2. As for the problem of more tables, we have added a table in section 3 and section 4 respectively to summary the calcium signaling pathways which are related to the occurrence of obesity.

    Figures and tables are attached to the submission.

 We tried our best to improve the manuscript and made many changes in the manuscript. These changes will not influence the content and framework of the paper. And here we did not list all the changes but marked in red in our revised paper.

We appreciate for reviewer#2' s warm work earnestly, and hope that the correction will meet with approval.

Once again, thank you very much for your comments and suggestions.

 Yours sincerely,

Ziguo Song

Name: Chao Sun

E-mail: sunchao2775@163.com

Round  2

Reviewer 1 Report

This version is improved, but requires extensive English editing.